# A comparative analysis of HIV status by sociodemographic and sexual behavior characteristics among men in Bexar county, Texas: An Ending the HIV Epidemic priority county

Adolph J. Delgado ⓘ *, Jeralynn S. Cossman, Rhonda BeLue

College for Health, Community and Policy, University of Texas at San Antonio, San Antonio, Texas, United States of America

* adolph.delgado2@utsa.edu

## Abstract

A combination of sociodemographic and sexual behaviors is associated with HIV status awareness. We collaborated with HIV service organizations to collect survey data from 389 men in Bexar County, Texas, an Ending the HIV Epidemic (EHE) priority county. We examined sociodemographic and behavioral characteristics across three self-reported HIV status groups: HIV-negative, HIV-positive, and unknown HIV status. Bivariate tests compared characteristics across the three groups, and multinomial logistic regression models assessed factors associated with HIV status, using HIV-negative and unknown status as alternating reference categories. Overall, 35% of respondents were HIV-negative, 26% were HIV-positive, and 39% did not know their HIV status. Compared with men who did not know their status, HIV-negative men were more likely to be non-Hispanic White, employed, married, and to have higher educational attainment and income. In multivariable models, higher education and employment were associated with greater odds of being HIV-negative relative to unknown status, whereas lower education and unemployment were associated with unknown status. Income differences were more nuanced, with men of unknown status more likely to report higher income categories than HIV-negative men. Men with unknown status reported lower condom use and less alcohol use during sex than HIV-negative men. Compared with men with unknown status, HIV-positive men were less likely to report recent casual and anal sex but more likely to report condom use and drug use during sex. These findings indicate that, in this EHE county, HIV status awareness among men is patterned by intersecting sociodemographic and behavioral characteristics rather than by single risk indicators. Men who know their HIV status, particularly those who are HIV-negative, more often occupy socioeconomically advantaged positions and report distinct sexual behavior profiles compared with men whose HIV status is unknown. Although these cross-sectional associations cannot

**Data availability statement:** The minimal de-identified dataset and accompanying codebook underlying the findings of this study are available as Supporting Information files and can be accessed as S1 Dataset and S2 Codebook with this article.

**Funding:** The author(s) received no specific funding for this work.

**Competing interests:** No authors have competing interests.

establish causality, they highlight men with lower education, unstable employment, and unknown status as a locally relevant, under-reached group for targeted HIV testing and outreach to advance Bexar County's EHE goals.

## Introduction

The Ending the HIV Epidemic (EHE) initiative aims to end the HIV epidemic in the United States through four science-based strategies: Diagnose, Treat, Prevent, and Respond [1,2]. These pillars collectively address the complex challenges of the HIV epidemic with a coordinated national response. The Diagnose pillar prioritizes early detection of HIV in all individuals to facilitate rapid and effective treatment, sustaining viral suppression and preventing new infections via treatment-as-prevention (TasP) [1–3]. This pillar also enhances the prevention of new transmissions through interventions such as pre-exposure prophylaxis (PrEP) and syringe service programs (SSPs) by expediting responses to HIV cluster detections and outbreaks and ensuring access to prevention and treatment resources [1,2,4,5].

The importance of the *Diagnose* pillar is evident in the efforts across various states to reduce new HIV transmissions, particularly in southern states, including Texas, where HIV transmission remains a significant public health concern [1,2,6–8]. From 2017 to 2022, the percentage of Texans living with HIV who knew their HIV status increased from 82% to 84% [8]. However, Texas estimate remains slightly below recent national estimates, which indicate that about 87% of people living with HIV in the United States know their status [9].

In Texas, five EHE priority counties—areas where more than 50% of new HIV diagnoses occurred in 2016 and 2017—are intensively targeted for enhanced interventions [1,7,8]. In one such county, 16 HIV molecular clusters were identified in 2017, with the largest cluster located in San Antonio, the most populous city in Bexar County [10,11].

The Texas Department of State Health Services (DSHS) has made significant strides in increasing HIV awareness through targeted testing programs at community-based organizations and by expanding routine HIV testing in areas with limited access to quality healthcare [8]. The strategic focus is on populations at a higher risk of HIV by addressing risky sexual behaviors such as casual encounters, *hooking up*, and the use of substances and alcohol during sexual activities, which are associated with increased HIV risk [12–19]. Additionally, the strategy aims to enhance HIV status awareness by extending testing services to communities characterized by low income, lower educational attainment, and high unemployment rates, socioeconomic determinants that often influence access to healthcare [8,20,21].

Bexar County, the fourth most populous county in Texas and a key region in South Texas, has over 2 million residents, within the San Antonio Metropolitan Area: a majority-minority area. Residents of Bexar County experience structural barriers, including limited access to high-quality education, lower household incomes, and a large low-wage workforce. These conditions can constrain access to timely, routine healthcare and contribute to delays in seeking care or receiving treatment when it is most effective. [11,13,22,23].

Approximately 14–15% of adults 25 years and older in Bexar County have not completed high school, reflecting lower educational attainment than national averages [24]. Nationally, workers of color ages 25–64 earn a median hourly wage of about $23, compared with $29 for white workers: a difference that amounts to roughly $12,480 less per year for full-time employment [25]. In Bexar County, about 15% of residents live below the federal poverty line, and a much larger share of households earn too much to qualify for public assistance yet too little to consistently meet basic needs [26]. Taken together, local assessments indicate that nearly half of households experience financial strain and have limited capacity to absorb unexpected expenses or build savings [27]. These socioeconomic disparities, coupled with the State of Texas's decision not to expand Medicaid, have contributed to persistent coverage gaps that hinder HIV testing, linkage to care, and HIV status awareness in communities such as Bexar County [28]. Evidence from non-expansion states shows that low-income adults are more likely to be uninsured and have limited access to routine health services, including HIV prevention and treatment. In Bexar County, the proportion of people living with HIV who know their status has remained in the low-80% range in recent years, with approximately 83–84% diagnosed: below national estimates of about 87% [29–31].

Given the EHE's targeted goals, AIDS services organizations (ASOs) and HIV prevention providers in Bexar County could benefit from understanding the sociodemographic and behavioral characteristics of individuals who know their HIV status and test HIV-negative, as well as those who do not know their status [32,33]. Individuals who are non-White, married, and have higher levels of education and income are more likely to be HIV-negative [34]. HIV-negative men frequently report higher rates of condomless sex and substance use [32,34]. Younger, Black, and single men who report sex with men are significantly more likely to have unknown HIV status compared with men who are HIV-negative or HIV-positive [33,34]. Men with less than a high school education have markedly higher odds of reporting unknown HIV status [33]. Men with unknown status who report sex with men differ from HIV-negative and HIV-positive men in their sexual behavior patterns, including more casual male partners in the past three months and more condomless anal insertive sex with same-sex partners [32].

Studies examining the influence of sociodemographic and behavioral characteristics on HIV status have identified distinct patterns of risk and prevention behavior across three groups: HIV-negative, HIV-positive aware, and HIV-positive unaware [32,33,35]. These findings offer valuable insights for HIV prevention providers seeking to engage HIV-unknown men in testing and routine sexual healthcare [32,33]. Our goal is to support Bexar County's EHE efforts by using empirical data to identify key factors influencing HIV status among men in a high-priority area. HIV providers can leverage these insights to allocate resources more effectively and implement targeted testing strategies based on the characteristics associated with HIV-negative men and those unaware of their status [32,33,35]. Aligning testing efforts with these factors can improve outreach, increase testing uptake, and enhance HIV status awareness.

## Materials and methods

### Study design and data collection

We conducted a cross-sectional online survey in collaboration with HIV service organizations in Bexar County, Texas, an Ending the HIV Epidemic (EHE) priority county. Participants were recruited from 22/06/2023 to 14/08/2023 through established HIV service delivery networks. The Alamo Area Resource Center (AARC), one of the oldest and most comprehensive professionally managed HIV service organizations in the region, disseminated recruitment materials directly to its active clients through case managers, clinic-based staff, and PrEP navigation services, providing clients the opportunity to participate during routine service encounters. Additional recruitment occurred through targeted email distribution via University Health (UHS), the primary administrative agency and key service provider for the federal Ryan White HIV/AIDS Program in Bexar County. UHS coordinates HIV care and support services across multiple provider organizations in the county, including the region's largest outpatient HIV clinical care settings. Recruitment materials included a brief description of the study, eligibility criteria, and a link to the Qualtrics survey.

To be eligible, individuals had to be at least 18 years old and provide informed consent after reading the electronic consent form. After consent, participants were asked about their HIV testing history and current HIV status. To meet eligibility criteria, respondents had to answer the HIV status item, which asked them to select whether they currently considered themselves HIV-negative, HIV-positive, or unsure/did not know their HIV status. HIV status for this analysis is based entirely on self-report from this item; we did not verify responses with medical records or laboratory data. The average survey completion time was 34 minutes, and participants were entered into a raffle to win a $25 Amazon gift card. For this analysis, only data from participants whose sex at birth was male and who self-identified as male were included ($n = 389$). The survey instrument was designed to capture sexual and substance-use behaviors that are central to HIV acquisition and HIV status awareness among men in this local context. Given this behavioral focus and the fact that our community partners primarily serve men at elevated HIV risk, we restricted the analytic sample to participants whose sex assigned at birth was male and who self-identified as male. The survey was available only in English. The study was approved by the UT San Antonio Institutional Review Board (FY22-23-42).

## Measures

**HIV Status.** Participants' HIV status was determined through a two-step self-reported process. First, they were asked, "Do you know your HIV status?" Those who answered Yes, confirming awareness of their testing history, were further asked, "What is your current HIV status?" Responses categorized participants as HIV Negative if they reported being negative or HIV Positive if they reported being positive. Participants who answered No to the initial question, indicating unawareness of their testing history, were categorized as Does Not Know.

## Sociodemographic characteristics

Participants' sociodemographic characteristics were categorized as follows: Race and ethnicity were classified as Hispanic/Latino, Non-Hispanic Black, or Non-Hispanic White. Due to small sample sizes, Hispanic/Latino and Black participants were combined into a single category termed Black, Indigenous, and People of Color (BIPOC). Educational attainment was grouped into six levels: less than high school, high school graduate, technical degree, some college, undergraduate degree, or graduate degree. Due to small cell sizes, income was categorized into five groups: Low Income (<$30,000), Middle Income ($30,000–$59,999), Upper Middle Income ($60,000–$89,999), High Income ($90,000–$149,999), and Very High Income (≥$150,000). Employment status was classified as either Unemployed or Employed, while marital status was categorized as Single or Married. Age was initially grouped into six categories (18–24, 25–34, 35–44, 45–54, 55–64, 65–74, and 75–84); however, due to small cell sizes, these were recategorized as Young Adult (18–34), Middle-Aged (35–54), and Senior (≥55).

## Sexual behaviors

Participants' sexual behaviors were assessed using seven questions. The first asked, "Have you had sex with another person(s) in the last 30 days?" with response options of No or Yes. The second question, "Who have you had sex (anal, oral) with in the last 6 months?" offered options of Men, Women, or Both Men and Women. To align with study definitions, responses were adjusted to form the Men Who Have Sex with Men (MSM) group. The third question, "Have you had casual sex with a person you just met and are not in a relationship with ('hooked up') in the last 6 months?" had response options of No or Yes. The fourth question, "Have you ever had anal sex?" categorized responses as Yes, receptive (bottoming); Yes, penetrative (topping); Yes, both receptive and penetrative; or No. Condom use was assessed with "How often do you use condoms during sex?" with responses ranging from Never to Always. The sixth and seventh questions evaluated substance use and alcohol consumption during sex, respectively. Participants were asked, "How often have you taken drugs/substances (e.g., poppers) during sex?" and "How often have you been drunk from alcohol during sex?" both with response options ranging from Never to Always.

## Statistical analysis

We examined characteristics associated with HIV status by conducting Chi-square tests for categorical variables and Kruskal-Wallis tests for ordinal variables. These tests compared sociodemographic and behavioral characteristics across the three HIV status groups: HIV-negative, HIV-positive, and Does Not Know (unknown status). We then used Bonferroni-adjusted post hoc tests to identify specific group differences and carried forward only variables with statistically significant associations ($p < 0.05$) for further analysis. We then ran two multinomial logistic regression models on the multiply imputed datasets. In the first model, we used HIV-negative participants as the reference group since they knew their status and tested negative, making them the gold standard for comparison. In the second model, we used participants with unknown HIV status as the reference group to identify characteristics that may act as barriers to testing.

## Missing data

We conducted Little's MCAR test to determine the randomness of the missing data, which returned a chi-square statistic of 58.37 with 41 degrees of freedom and was significant at $p = 0.04$. This result led us to reject the null hypothesis of data being missing completely at random (MCAR), confirming our suspicion of a missing at random (MAR) pattern influenced by observed variables in the dataset. Race was identified as having the highest percentage of missing data at 13.9%, underscoring potential challenges in capturing racial demographics. Conversely, education had the lowest percentage of missing data at 0.3%. The presence of systematic missing data underscored the necessity for robust missing data techniques. Multiple imputation via the Fully Conditional Specification (FCS) method was used to handle the missing data for sociodemographic and sexual behavior variables. The FCS method was employed with a maximum of 50 iterations and 5 imputations per missing instance. This method was chosen due to its flexibility in handling different types of missing data. Predictive mean matching (PMM) was specifically used for scale variables to enhance the accuracy of imputation by matching imputed values with observed values that are closest in the dataset. We pooled results from five imputations to ensure robust descriptive statistics to account for variability across datasets, yielding outcomes that closely approximate true values. The total number of observations (N = 389) remained consistent across all imputed datasets, reflecting comprehensive study coverage. All imputations and computations were conducted using SPSS version 29 [36].

## Results

### Bivariate analyses

The study included 389 participants: 35% (n = 135) were HIV-negative, 26% (n = 102) were HIV-positive, and 39% (n = 152) had unknown HIV status. **Table 1** presents sociodemographic differences across HIV status groups. Race distributions differed significantly by HIV status ($\chi^2 = 9.16$, $p = .010$), with a higher proportion of BIPOC respondents among HIV-positive men (49%) compared with HIV-negative men (30%). Employment also differed by HIV status ($\chi^2 = 11.76$, $p = .003$); unemployment was most common among men who did not know their HIV status (24%) compared with HIV-negative men (9%). Marital status varied significantly across groups ($\chi^2 = 20.01$, $p < .001$), with HIV-negative men more likely to be married (67%) than HIV-positive men (42%) and those who did not know their status (45%). Age group did not differ significantly by HIV status (H = 2.11, $p = .349$).

Education levels differed substantially across HIV status groups (H = 44.87, $p < .001$). HIV-negative men reported the highest education level on average (Mdn = 4, IQR = 2–4), corresponding to an undergraduate degree, while HIV-positive men reported lower education (Mdn = 2, IQR = 1–3), corresponding to a technical degree, and men who did not know their status reported intermediate levels (Mdn = 3, IQR = 2–4), corresponding to some college. Income also differed across groups (H = 10.97, p = .004), with HIV-positive men showing a lower income distribution (Mdn = 2, IQR = 1–2; middle income) relative to men who did not know their status (Mdn = 2, IQR = 2–3; middle to upper-middle income), and HIV-negative men (Mdn = 2, IQR = 1–3; middle to upper-middle income).

**Table 1. Sociodemographic characteristics by self-reported HIV status (HIV-negative, HIV-positive, unknown).**

| Variable | Category | HIV Negative (n = 135)[A] | | HIV Positive (n = 102)[B] | | Does Not Know (n = 152)[C] | | Test (χ²) | p | Post-hoc |
|---|---|---|---|---|---|---|---|---|---|---|
| | | n | % | n | % | n | % | | | |
| Race | BIPOC | 41 | 30% | 50 | 49% | 53 | 35% | 9.16 | 0.010 | A≠B |
| | NH-White | 94 | 70% | 52 | 51% | 99 | 65% | | | |
| Employ-ment | Unemployed | 12 | 9% | 15 | 15% | 36 | 24% | 11.76 | 0.003 | A≠C |
| | Employed | 123 | 91% | 87 | 85% | 116 | 76% | | | |
| Marital Status | Single | 44 | 33% | 59 | 58% | 84 | 55% | 20.01 | <0.001 | A≠B; A≠C |
| | Married | 91 | 67% | 43 | 42% | 68 | 45% | | | |
| | | Mdn | IQR | Mdn | IQR | Mdn | IQR | H | | |
| Age | Age group (1 = 18–34, 2 = 35–54, 3 = 55+) | 1 | 1-2 | 1 | 1-2 | 1 | 1-2 | 2.11 | 0.349 | |
| Education | Ordered categories (0 = Less than HS … 5 = Graduate) | 4 | 2-4 | 2 | 1-3 | 3 | 2-4 | 44.87 | <0.001 | A≠B; A≠C; B≠C |
| Income | Ordered categories (1=<$30k … 5=>$150k) | 2 | 1-3 | 2 | 1-2 | 2 | 2-3 | 10.97 | 0.004 | A≠B; B≠C |

Note. Percentages are rounded to the nearest whole number. Age, education, and income are treated as ordinal variables; values shown for these rows are the median (Mdn) and interquartile range (IQR). Chi-square (χ²) tests were used for categorical variables; Kruskal–Wallis H tests were used for ordinal variables.

Table 2 presents behavioral differences by HIV status. Sexual activity differed significantly across groups (χ² = 55.90, p < .001): 84% of HIV-negative men reported being sexually active, compared with 46% of HIV-positive men and 45% of men who did not know their status. MSM behavior did not differ significantly by HIV status (χ² = 3.79, p = .151). Casual sex differed across groups (χ² = 9.77, p = .008), with the highest prevalence among HIV-positive men (65%), followed by HIV-negative men (53%) and men who did not know their status (45%). Anal sex also varied significantly (χ² = 12.38, p = .002), reported by 60% of HIV-negative men and 61% of HIV-positive men compared with 42% of men who did not know their status. Condom use differed significantly across HIV status groups (H = 16.60, p < .001), with HIV-positive men reporting more frequent condom use (Mdn = 2, IQR = 1–3; about half the time) compared with HIV-negative men (Mdn = 1, IQR = 1–3; sometimes) and men who did not know their status (Mdn = 1, IQR = 1–2; sometimes). Drug use also differed significantly (H = 10.29, p = .006), with HIV-positive men reporting higher levels and greater variability in drug use during sex (Mdn = 1, IQR = 1–3) compared with HIV-negative men (Mdn = 1, IQR = 0–2) and those who did not know their status (Mdn = 1, IQR = 1–1). Alcohol use differed significantly as well (H = 10.98, p = .004), with HIV-negative men reporting slightly higher and more variable alcohol use during sex (Mdn = 1, IQR = 1–2) compared with HIV-positive men (Mdn = 1, IQR = 0–1) and men who did not know their status (Mdn = 1, IQR = 1–1).

## Multivariable analyses

Table 3 presents the results of a multinomial logistic regression comparing sociodemographic and behavioral characteristics between men who are HIV-positive or whose HIV status is unknown, with HIV-negative men as the reference category. Relative to HIV-negative men, HIV-positive men had significantly lower odds of higher educational attainment (OR = 0.57, p < .001). HIV-positive men also had significantly higher odds of reporting that they were not sexually active (OR = 5.43, p < .001). In addition, HIV-positive men had significantly lower odds of reporting no anal sex, indicating higher odds of reporting anal sex (OR = 0.48, p = .046). No other sociodemographic or behavioral variables were statistically significant in this comparison.

**Table 2. Sexual and substance use behaviors by self-reported HIV status (HIV-negative, HIV-positive, unknown).**

| Variable | Category | HIV Negative[A] n=135 | | HIV Positive[B] n=102 | | Does Not Know[C] n=152 | | χ² | p | Post-Hoc |
|---|---|---|---|---|---|---|---|---|---|---|
| | | n | % | n | % | n | % | | | |
| Sexually Active | No | 21 | 16% | 55 | 54% | 84 | 55% | 55.90 | <0.001 | A≠B |
| | Yes | 114 | 84% | 47 | 46% | 68 | 45% | | | |
| MSM Behavior | No | 88 | 65% | 72 | 71% | 115 | 76% | 3.79 | 0.151 | |
| | Yes | 47 | 35% | 30 | 29% | 37 | 24% | | | |
| Casual Sex | No | 64 | 47% | 36 | 35% | 84 | 55% | 9.77 | 0.008 | B≠C |
| | Yes | 71 | 53% | 66 | 65% | 68 | 45% | | | |
| Anal Sex | No | 54 | 40% | 40 | 39% | 88 | 58% | 12.38 | 0.002 | A≠C, B≠C |
| | Yes | 81 | 60% | 62 | 61% | 64 | 42% | | | |
| Condom Use | | Mdn | IQR | Mdn | IQR | Mdn | IQR | H | | |
| | Never | 1 | 1-3 | 2 | 1-3 | 1 | 1-2 | | | |
| | Sometimes | | | | | | | 16.60 | <0.001 | A≠C, B≠C |
| | About half the time | | | | | | | | | |
| | Most of the time | | | | | | | | | |
| | Always | | | | | | | | | |
| Drug Use | Never | 1 | 0-2 | 1 | 1-3 | 1 | 1−1 | 10.29 | 0.006 | A≠B, B≠C |
| | Sometimes | | | | | | | | | |
| | About half the time | | | | | | | | | |
| | Most of the time | | | | | | | | | |
| | Always | | | | | | | | | |
| Alcohol Use | Never | 1 | 1-2 | 1 | 0-1 | 1 | 1−1 | 10.98 | 0.004 | A≠B, A≠C |
| | Sometimes | | | | | | | | | |
| | About half the time | | | | | | | | | |
| | Most of the time | | | | | | | | | |
| | Always | | | | | | | | | |

Note: Percentages are rounded to the nearest whole number. Condom use, drug use, and alcohol use are modeled as ordinal variables, with coefficients reflecting changes between ordered categories. An independent samples Kruskal-Wallis test was used to interpret the differences between these categories.

Men with unknown HIV status had significantly higher odds of being unemployed (OR = 4.11, $p = .004$), lower odds of higher educational attainment (OR = 0.73, $p = .010$), and higher odds of being in a higher income category (OR = 1.44, $p = .023$). Men with unknown HIV status also had significantly higher odds of reporting that they were not sexually active (OR = 5.73, p < .001). They had lower odds of more frequent condom use (OR = 0.69, $p = .007$) and lower odds of more frequent alcohol use during sex (OR = 0.60, $p = .003$).

Table 4 presents a multinomial logistic regression comparing men who know their HIV status (HIV-negative or HIV-positive) with those who do not know their status (reference category). Compared with men who did not know their HIV status, HIV-negative men had significantly lower odds of being unemployed (OR = 0.25, $p = .002$) and significantly lower odds of being single (i.e., higher odds of being married; OR = 0.46, $p = .012$). HIV-negative men also had higher odds of higher educational attainment (OR = 1.41, $p = .004$) and lower odds of being in a higher income category (OR = 0.73, $p = .038$). In addition, HIV-negative men had lower odds of reporting no anal sex, indicating higher odds of reporting anal sex (OR = 0.53, $p = .036$). HIV-negative men also had higher odds of more frequent condom use (OR = 1.39, $p = .010$) and higher odds of more frequent alcohol use during sex (OR = 1.65, $p = .002$).

**Table 3. Comparing sociodemographic and behavioral characteristics by people who are HIV positive or whose HIV status is unknown to those who are HIV negative.**

| Outcome | Predictor | OR | Lower | Upper | p |
|---|---|---|---|---|---|
| HIV Positive[B] | Intercept | | | | 0.215 |
| | Race: BIPOC (ref: NH-White) | 1.66 | 0.83 | 3.29 | 0.150 |
| | Employment: Unemployed (ref: Employed) | 0.83 | 0.30 | 2.30 | 0.717 |
| | Marital status: Single (ref: Married) | 1.50 | 0.72 | 3.10 | 0.278 |
| | Education | 0.57 | 0.44 | 0.73 | 0.000 |
| | Income | 0.85 | 0.60 | 1.22 | 0.385 |
| | Sexually active: No (ref: Yes) | 5.43 | 2.74 | 10.76 | 0.000 |
| | Anal sex: No (ref: Yes) | 0.48 | 0.23 | 0.99 | 0.046 |
| | Condom Use | 1.21 | 0.91 | 1.62 | 0.187 |
| | Drug Use | 1.14 | 0.86 | 1.49 | 0.363 |
| | Alcohol Use | 0.72 | 0.51 | 1.02 | 0.064 |
| Does Not Know[C] | Intercept | | | | 0.343 |
| | Race: BIPOC (ref: NH-White) | 0.69 | 0.36 | 1.34 | 0.279 |
| | Employment: Unemployed (ref: Employed) | 4.11 | 1.57 | 10.76 | 0.004 |
| | Marital status: Single (ref: Married) | 1.73 | 0.89 | 3.37 | 0.106 |
| | Education | 0.73 | 0.58 | 0.93 | 0.010 |
| | Income | 1.44 | 1.05 | 1.96 | 0.023 |
| | Sexually active: No (ref: Yes) | 5.73 | 3.07 | 10.67 | 0.000 |
| | Anal sex: No (ref: Yes) | 1.46 | 0.78 | 2.73 | 0.240 |
| | Condom Use | 0.69 | 0.53 | 0.90 | 0.007 |
| | Drug Use | 0.81 | 0.61 | 1.07 | 0.141 |
| | Alcohol Use | 0.60 | 0.43 | 0.84 | 0.003 |

Note: The reference category is HIV Negative[A].

In contrast, compared with men who did not know their HIV status, HIV-positive men had significantly lower odds of being unemployed (OR = 0.24, *p* = .001) and lower odds of being in a higher income category (OR = 0.52, *p* < .001). HIV-positive men also had lower odds of reporting no casual sex (indicating higher odds of reporting casual sex; OR = 0.39, *p* = .004) and lower odds of reporting no anal sex (indicating higher odds of reporting anal sex; OR = 0.39, *p* = .005). HIV-positive men had higher odds of more frequent condom use (OR = 1.85, *p* < .001) and higher odds of more frequent drug use during sex (OR = 1.38, *p* = .023). No significant differences were observed for marital status, education, or alcohol use during sex in this comparison.

## Discussion

Using data collected from 389 male respondents in Bexar County, Texas, we found that HIV status awareness among men was patterned by both sociodemographic and behavioral characteristics [13,20,21]. Men who were HIV-negative were significantly more likely than men with positive or unknown status to be non-Hispanic White, married, employed, and to have completed higher levels of education. These patterns are consistent with broader evidence that educational attainment, stable employment, and higher income enable more regular engagement with healthcare systems, better access to preventive services, and greater opportunities to receive HIV testing [21]. In contrast, men who did not know their HIV status experienced greater socioeconomic disadvantage on employment and education than HIV-negative men. Taken together, these findings echo prior work linking economic hardship and limited educational opportunity to reduced HIV

**Table 4. Comparing sociodemographic and behavioral characteristics by people who know their HIV status (positive or negative) to those who do not know.**

| Outcome | Predictor | OR | Lower | Upper | p |
|---|---|---|---|---|---|
| HIV Negative[A] | Intercept | | | | 0.295 |
| | Employment: Unemployed (ref: Employed) | 0.25 | 0.10 | 0.61 | 0.002 |
| | Marital status: Single (ref: Married) | 0.46 | 0.25 | 0.84 | 0.012 |
| | Education | 1.41 | 1.12 | 1.79 | 0.004 |
| | Income | 0.73 | 0.54 | 0.98 | 0.038 |
| | Casual sex: No (ref: Yes) | 0.60 | 0.34 | 1.04 | 0.070 |
| | Anal sex: No (ref: Yes) | 0.53 | 0.30 | 0.96 | 0.036 |
| | Condom Use | 1.39 | 1.08 | 1.80 | 0.010 |
| | Drug Use | 1.20 | 0.91 | 1.57 | 0.197 |
| | Alcohol Use | 1.65 | 1.20 | 2.28 | 0.002 |
| HIV Positive[B] | Intercept | | | | 0.119 |
| | Employment: Unemployed (ref: Employed) | 0.238 | 0.100 | 0.568 | 0.001 |
| | Marital status: Single (ref: Married) | 0.858 | 0.438 | 1.681 | 0.656 |
| | Education | 0.830 | 0.649 | 1.061 | 0.136 |
| | Income | 0.522 | 0.368 | 0.741 | 0.000 |
| | Casual sex: No (ref: Yes) | 0.385 | 0.199 | 0.743 | 0.004 |
| | Anal sex: No (ref: Yes) | 0.386 | 0.199 | 0.747 | 0.005 |
| | Condom Use | 1.854 | 1.390 | 2.472 | 0.000 |
| | Drug Use | 1.383 | 1.046 | 1.829 | 0.023 |
| | Alcohol Use | 1.087 | 0.752 | 1.571 | 0.658 |

Note: The reference category is Does Not Know[C].

testing and delayed status awareness and suggest that structural barriers, not merely individual behavior, can influence who accesses HIV testing and learns their status.

At the same time, HIV-negative men in this sample reported higher rates of sexual activity and, descriptively, more sex with other men than men in the unknown-status group. On the surface, this appears to contradict the expectation that higher levels of condomless sex and substance use necessarily translate into higher HIV prevalence [15,16,19,32,33]. However, there are several plausible explanations that are consistent with existing prevention science. First, HIV-negative men with higher education and more stable employment may have greater access to HIV testing, PrEP, condoms, and sexual health counseling, allowing them to engage in risk-reduction strategies even while remaining sexually active. Second, these men may be more likely to seek regular care in primary care or sexual health settings where routine testing is offered, which increases the likelihood of detecting HIV early or confirming that they remain negative [13,20,21,35]. Third, biological and temporal factors, such as recent exposure, window periods, or differential timing of testing, may mean that some currently negative men could seroconvert later; our cross-sectional design cannot capture such dynamics [3,4,6,30]. For these reasons, these findings should not be interpreted as evidence that high-risk sexual behaviors are "safe" for HIV-negative men, but rather as an indication that risk and protection operate simultaneously and are strongly conditioned by access to structural and biomedical resources.

Comparisons between men living with HIV and those with unknown status further underscore the importance of structural and programmatic factors. Both groups showed indicators of socioeconomic vulnerability, including lower education relative to HIV-negative men, and men with unknown status had the highest levels of unemployment [20,21,28,30,37]. Men reporting no recent sexual activity or lower engagement in recognized risk behaviors were more likely to have

unknown HIV status, suggesting that risk-based testing strategies may systematically overlook men who are sexually inactive, intermittently active, or who do not self-identify as being at high risk [21,32,33].

This is particularly salient in Bexar County, where EHE efforts are often implemented through HIV-focused organizations, clinics, and outreach venues that may not routinely reach men with lower perceived risk. Men living with HIV, by contrast, were more likely to report casual and anal sex but also higher condom use, a pattern compatible with greater historical exposure that led to diagnosis and subsequent access to risk-reduction counseling and prevention resources. Once diagnosed, individuals may receive repeated messages about safer sex, partner notification, and viral suppression, which can alter behavior over time in ways that are not captured by a simple cross-sectional contrast.

Prevention and compensatory behaviors also varied by HIV status. Higher condom use among men living with HIV compared with those with unknown status likely reflects prevention counseling and education that accompany diagnosis, as well as personal motivation to reduce onward transmission to partners. Elevated drug use during sex among men living with HIV may indicate that they are embedded in social and sexual networks where substance use and HIV risk overlap and where targeted outreach, testing, and treatment services are more commonly delivered. In contrast, men with unknown status may have fewer contacts with these specialized networks and services, leaving both their substance use and HIV risk unaddressed. Although these associations cannot establish causal pathways, they help differentiate men who are already connected to HIV prevention and care systems from those who remain on the periphery. From a prevention standpoint, these patterns suggest that status-neutral interventions that target substance-using networks, casual partner venues, and online spaces may be effective not only for men living with HIV but also for those whose status is unknown [1,2,20,33,38].

Taken together, these findings indicate that HIV status in Bexar County reflects an interplay of demographic, economic, and behavioral factors rather than a single, easily defined "risk profile" [21,32,33]. The primary contribution of this study is not to identify novel behavioral risk factors, but to provide county-specific evidence from an Ending the HIV Epidemic priority jurisdiction and to characterize men with unknown HIV status as a distinct, under-reached group within a service-engaged male population. In a context where EHE benchmarks emphasize increasing the proportion of people who know their status, our results suggest that focusing solely on men who present to traditional HIV testing venues or who identify as "high risk" will miss a substantial subset of men facing structural disadvantage, intermittent sexual activity, or lower perceived risk [15,16,35].

For public health practice, these results point toward several concrete implications for Bexar County's EHE implementation. First, testing strategies should be expanded beyond specialized HIV clinics and community events that primarily reach men who already identify as at risk or who are embedded in HIV-related networks. Integrating status-neutral HIV testing into primary care practices, urgent care settings, emergency departments, workforce development programs, and other social service settings could help reach men who are unemployed, underemployed, or less connected to traditional HIV services. Second, collaborations between AIDS service organizations, Ryan White–funded programs, and agencies focused on employment, housing, and education could facilitate co-located services that address both HIV risk and the structural determinants that hinder testing. Third, communication strategies should explicitly target men who do not see themselves as "high risk," including those with infrequent or no recent sexual activity, emphasizing the benefits of routine HIV screening as a standard component of adult preventive care rather than a marker of deviance or stigma.

Finally, these findings can inform local monitoring and evaluation efforts within the EHE framework. By identifying specific combinations of demographic characteristics (e.g., unemployment, lower education), behavioral patterns (e.g., low condom use, low testing history), and service engagement (e.g., limited contact with HIV programs) that characterize men with unknown status, Bexar County stakeholders can refine outreach, prioritize neighborhoods or networks for intensified testing, and track whether status awareness improves over time in these groups. Although the cross-sectional design and non-probability sampling limit causal inference and generalizability, the patterns documented here are useful for hypothesis generation and for tailoring local interventions. HIV testing in Bexar County must be decoupled from behavioral risk categories and instead integrated into economic and social services such as workforce development programs, primary

care clinics, and financial assistance services. Only by addressing the structural conditions that limit testing access, unemployment, low education, healthcare distrust, can Bexar County move toward equitable status awareness and the EHE 2030 goal of 95% knowledge of HIV status [1,2,7,30].

## Limitations and strengths

The study design presents several inherent limitations. First, as a cross-sectional online survey, it captures only a snapshot of behaviors (e.g., casual sex, substance use) and characteristics (e.g., employment) at the time participants completed the questionnaire. This limits our ability to establish temporal ordering or make causal inferences, and all findings should be interpreted as correlational rather than causal [39]. Second, our method of recruitment may introduce selection bias. Participants were recruited through the Alamo Area Resource Center (AARC) and the San Antonio Ryan White Program's social media and email networks, which likely attracted individuals who are connected to HIV service organizations or who are more comfortable engaging with health-related content online. While this approach facilitated participation among groups who can be difficult to reach due to stigma and other barriers, it may underrepresent men who are not engaged in services or who are less connected to these networks [40,41]. Third, we did not ask participants who knew their HIV status when they were last tested or diagnosed. Without this information, we cannot determine how recently participants learned of their status or whether awareness is long-standing or recent, which may influence current behaviors and responses. Future studies should include detailed testing history items to provide a more comprehensive understanding of how timing of testing and diagnosis relates to HIV status awareness and behavior over time. Fourth, use of an anonymous online survey has implications for data quality. Although the Qualtrics platform recorded completion time and response patterns for each case and we examined these metadata for implausibly short completion times, highly patterned response sets, and obviously duplicated entries, no responses met our criteria for exclusion and all eligible cases were retained. We did not implement additional automated tools such as CAPTCHA or IP-based blocking to prevent multiple or automated entries, and it remains possible that some undetected fraudulent or duplicate responses are present. This is a general limitation of anonymous online survey research and should be considered when interpreting the findings [42,43]. Fifth, reliance on self-reported data introduces the potential for recall error and social desirability bias. Participants may have underreported behaviors perceived as risky (e.g., condomless sex, drug use during sex) or overreported protective behaviors (e.g., consistent condom use), which could bias estimates of the prevalence of specific behaviors and their associations with HIV status categories [44]. In addition, the survey was conducted exclusively in English, which may have excluded men with limited English proficiency. Although many San Antonio residents speak only English at home, future research should incorporate bilingual or Spanish-language instruments to better capture the experiences of linguistically diverse populations [45]. Finally, our sample consisted primarily of non-Hispanic White men (63%), whereas Bexar County's adult population is majority Latino. This discrepancy reflects our recruitment through HIV service organizations and an online, English-only survey, which may have preferentially reached non-Hispanic White and more service-connected participants. As a result, the sample is not representative of all men in Bexar County, particularly Latino men, and the findings should be interpreted as describing a convenience sample of men engaged with or reachable through these channels rather than the county's male population as a whole. Future studies should employ targeted recruitment strategies or over-sampling to ensure adequate representation of Latino and other underrepresented groups.

## Supporting information

**S1 Dataset. Deidentified dataset underlying the findings of this study.**
(CSV)

**S2 File. Variable codebook for the dataset.**
(PDF)

## Acknowledgments

The authors would like to extend their deepest gratitude to the Alamo Area Resource Center (AARC) and the San Antonio Ryan White Program for their invaluable support and collaboration in this study.

## Author contributions

**Conceptualization:** Jeralynn S. Cossman, Rhonda BeLue.

**Data curation:** Adolph Delgado.

**Formal analysis:** Adolph Delgado.

**Methodology:** Adolph Delgado.

**Supervision:** Jeralynn S. Cossman, Rhonda BeLue.

**Writing – original draft:** Adolph Delgado.

**Writing – review & editing:** Jeralynn S. Cossman, Rhonda BeLue.

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
