## [Decision Letter · Decision Letter 0]

17 Oct 2025

PONE-D-25-10308
A Comparative Analysis of HIV Status by Sociodemographic and Sexual Behavior Characteristics among Men in Bexar County, Texas: An Ending the HIV Epidemic (EHE) Targeted County
PLOS One

Dear Dr. Delgado,

Thank you for submitting your manuscript to PLOS ONE. After careful consideration, we feel that it has merit but does not fully meet PLOS ONE’s publication criteria as it currently stands. Therefore, we invite you to submit a revised version of the manuscript that addresses the points raised during the review process.
 
Please note that we have only been able to secure a single reviewer to assess your manuscript. We are issuing a decision on your manuscript at this point to prevent further delays in the evaluation of your manuscript. Please be aware that the editor who handles your revised manuscript might find it necessary to invite additional reviewers to assess this work once the revised manuscript is submitted. However, we will aim to proceed on the basis of this single review if possible.

The reviewer has raised a number of points that need addressing - their comments are available below. They feel the interpretation and discussion of the study's findings should be reframed, and have questions about the analysis methods and quality of the data.

Could you please carefully revise the manuscript to address all comments raised?

We look forward to receiving your revised manuscript.

Sincerely,

Alejandro Torrado Pacheco, PhD

Associate Editor

PLOS One

Journal Requirements:

No authors have competing interests

Reviewers' comments:

Reviewer's Responses to Questions

**Comments to the Author**

1. Is the manuscript technically sound, and do the data support the conclusions?

Reviewer #1: Yes

2. Has the statistical analysis been performed appropriately and rigorously?

Reviewer #1: Yes

3. Have the authors made all data underlying the findings in their manuscript fully available?

Reviewer #1: No

4. Is the manuscript presented in an intelligible fashion and written in standard English?

Reviewer #1: Yes

5. Review Comments to the Author

Reviewer #1: This manuscript explores sociodemographic and behavioral correlates of HIV status awareness among men in Bexar County, Texas, an EHE priority jurisdiction. The topic is locally relevant, with implications for targeted HIV testing and prevention strategies. The paper is clearly well-written, and includes a substantial amount of contextual background on the EHE initiative and local epidemiology.

However, there are several areas that require further clarification, methodological refinement, and stronger interpretation to maximize the manuscript’s contribution.

1. Unfortunately the sample disproportionately non-Hispanic White (63%) compared to the actual demographic profile of Bexar County (majority Latino) and calls to question the representativeness of the findings for Bexar County

2. The conclusion that HIV-negative men engage in more high-risk behaviors yet remain HIV-negative due to education/economic stability is speculative. Biological and temporal factors (e.g., window periods, testing access) may also play roles. Greater caution is needed in these interpretations.

3. The study limits the analytic sample to men, but no clear justification is provided for this decision. While men who have sex with men are indeed a key population for HIV prevention, excluding women and gender-diverse individuals without explanation diminishes the broader relevance of the study and risks overlooking disparities in other groups in Bexar County.

4. The description of eligibility is unclear. The manuscript states that participants were eligible if they consented, were ≥18 years old, and “indicated their HIV status.” However, it is not defined what “indicating” HIV status entails

5. As the study relied on an online Qualtrics survey distributed via social media and email, procedures to detect and prevent fraudulent responses (e.g., bots, duplicate entries, ineligible participants) should be clearly described. The manuscript does not indicate whether tools such as CAPTCHA, IP address checks, or survey completion time thresholds were used. This omission raises concerns about data quality and the validity of the analytic sample.

6. The Discussion is lengthy and at times disorganized, with substantial repetition of results. The authors should reduce the regurgitation of descriptive findings and instead focus on the key implications for HIV prevention and public health policy.

7. Many of the associations (education, employment, condom use, income) are already well-established in the HIV literature. The true contributions of this study are: providing county-specific data in an EHE target area, and highlighting the characteristics of individuals with “unknown HIV status.”The manuscript should make this clear, rather than suggesting the findings themselves are novel.

6. PLOS authors have the option to publish the peer review history of their article (what does this mean?). If published, this will include your full peer review and any attached files.

Reviewer #1: No

---

## [Author Response · Author response to Decision Letter 1]

12 Dec 2025

We thank the Academic Editor and reviewer for their thoughtful and constructive feedback. We have revised the manuscript accordingly and provide a detailed, point-by-point “Response to Reviewers” document uploaded with this submission.

In brief, we: (1) clarified eligibility criteria and the rationale for the men-only analytic sample; (2) added a Data Quality Assurance subsection; (3) corrected and refined the Results text so that interpretations align precisely with Tables 1–4; (4) substantially condensed and reorganized the Discussion to reduce repetition and focus on implications for HIV prevention and EHE policy; (5) expanded our description of sample representativeness and limitations; and (6) updated the Competing Interests, Ethics, and Data Availability statements, and provided the de-identified analytic dataset and codebook as Supporting Information, in accordance with PLOS ONE policies.

---

## [Decision Letter · Decision Letter 1]

11 Feb 2026

A comparative analysis of HIV status by sociodemographic and sexual behavior characteristics among men in Bexar County, Texas: An Ending the HIV Epidemic priority county

PONE-D-25-10308R1

Dear Dr. Delgado,

We’re pleased to inform you that your manuscript has been judged scientifically suitable for publication and will be formally accepted for publication once it meets all outstanding technical requirements.

Kind regards,

Daniel Demant, PhD

Academic Editor

PLOS One

Additional Editor Comments (optional):

The authors have addressed all comments made by the reviewer adequately. While issues such as sampling bias were not fundamentally addresses, this is appropriate given this would involve the collection of new data. The re-framing is therefore appropriate and sufficient.

Reviewers' comments:

Reviewer's Responses to Questions

**Comments to the Author**

1. If the authors have adequately addressed your comments raised in a previous round of review and you feel that this manuscript is now acceptable for publication, you may indicate that here to bypass the “Comments to the Author” section, enter your conflict of interest statement in the “Confidential to Editor” section, and submit your "Accept" recommendation.

Reviewer #1: All comments have been addressed

2. Is the manuscript technically sound, and do the data support the conclusions?

Reviewer #1: Yes

3. Has the statistical analysis been performed appropriately and rigorously?

Reviewer #1: Yes

4. Have the authors made all data underlying the findings in their manuscript fully available?

Reviewer #1: No

5. Is the manuscript presented in an intelligible fashion and written in standard English?

Reviewer #1: Yes

6. Review Comments to the Author

Reviewer #1: The authors have addressed all of my comments satisfactorily. I have no other comments to add. The authors have addressed all of my comments satisfactorily. I have no other comments to add.

7. PLOS authors have the option to publish the peer review history of their article (what does this mean?). If published, this will include your full peer review and any attached files.

Reviewer #1: No

---

## [Editor Report · Acceptance letter]

PONE-D-25-10308R1

PLOS One

Dear Dr. Delgado,

I'm pleased to inform you that your manuscript has been deemed suitable for publication in PLOS One. Congratulations! Your manuscript is now being handed over to our production team.

Kind regards,

on behalf of

Associate Professor Daniel Demant

Academic Editor

PLOS One